# Endurance Training in Humans Modulates the Bacterial DNA Signature of Skeletal Muscle

**DOI:** 10.3390/biomedicines10010064

**Published:** 2021-12-29

**Authors:** Julia Villarroel, Ida Donkin, Camille Champion, Rémy Burcelin, Romain Barrès

**Affiliations:** 1Novo Nordisk Foundation Center for Basic Metabolic Research, Faculty of Health and Medical Sciences, University of Copenhagen, 2200 Copenhagen, Denmark; biotacg@gmail.com (J.V.); djx542@alumni.ku.dk (I.D.); 2Institut de Recherche sur les Maladies Métaboliques et Cardiovasculaires de Rangueil (I2MC), Institut National de la Santé et de la Recherche Médicale (INSERM), U1048, 31400 Toulouse, France; camille.champion@inserm.fr (C.C.); remy.burcelin@inserm.fr (R.B.); 3Institut des Mathématiques de Toulouse, 31400 Toulouse, France

**Keywords:** tissue-borne microbiome, 16S rRNA sequencing, skeletal muscle, endurance training

## Abstract

Accumulating evidence supports the existence of a tissue microbiota, which may regulate the physiological function of tissues in normal and pathological states. To gain insight into the regulation of tissue-borne bacteria in physiological conditions, we quantified and sequenced the 16S rRNA gene in aseptically collected skeletal muscle and blood samples from eight healthy male individuals subjected to six weeks of endurance training. Potential contamination bias was evaluated and the taxa profiles of each tissue were established. We detected bacterial DNA in skeletal muscle and blood, with background noise levels of detected bacterial DNA considerably lower in control versus tissue samples. In both muscle and blood, Proteobacteria, Actinobacteria, Firmicutes and Bacteroidetes were the most prominent phyla. Endurance training changed the content of resident bacterial DNA in skeletal muscle but not in blood, with *Pseudomonas* being less abundant, and both *Staphylococcus* and *Acinetobacter* being more abundant in muscle after exercise. Our results provide evidence that endurance training specifically remodels the bacterial DNA profile of skeletal muscle in healthy young men. Future investigations may shed light on the physiological impact, if any, of training-induced changes in bacterial DNA in skeletal muscle.

## 1. Background

The symbiosis between humans and bacteria has been long thought to be confined to bacteria resident on epithelia which is in direct contact with the external environment, such as the gastrointestinal tract, vagina, lungs, and skin. However, recently, we and others described the existence of a tissue microbiota in healthy and pathological situations, such as type 2 diabetes (T2D), in mice and humans [1,2,3].

Evidence of the existence of a healthy human microbiome in blood [4,5], breast [6,7,8], lung [9,10,11,12] and liver [13,14,15] is accumulating.

The origin of tissue-resident bacteria is unclear, but the finding that the gut-microbiota signature characteristic of metabolic diseases is also detectable in deep tissues, such as the adipose tissue and the liver, supports that bacteria translocate from the gut to these tissues [1,16]. An increase in permeability of the intestinal mucosa, as suggested in the leaky gut hypothesis, has been proven to be the mechanism at play in the translocation of bacteria to non-epithelial tissues [17,18,19,20,21,22]. Increased intestinal permeability is linked to the reduction of interleukin 17 (IL17) producing T-helper cells (Th17) in the lamina propria, resulting from the inability of antigen-presenting cells to differentiate Th17 cells [23].

Sequencing of the 16S ribosomal ribonucleic acid (rRNA) gene allows profiling of bacterial signatures in different tissues [24,25] and offers the potential to identify bacterial biomarkers indicative of specific metabolic states [1,3,23,26]. While numerous reports demonstrate the change of gut microbiota in exercise-trained individuals [27,28,29], they do not address the impact of lifestyle factors like endurance training in the control of tissue-resident bacteria. Here, we hypothesised that a lifestyle intervention like physical exercise training remodels the composition of blood and skeletal muscle-borne bacteria.

To assess the effect of endurance training on bacterial signatures in human peripheral tissues, we sequenced bacterial DNA content from blood and skeletal muscle, a primary exercise-effector tissue, as surrogate markers of the possible presence of tissue-borne bacteria. We performed a meticulous analysis of the bacterial DNA profiles of muscle and blood of healthy young men and a thorough evaluation of possible contamination levels. We explored the effect of a 6-week endurance training on bacterial communities and identified that exercise training impacts skeletal muscle, but not blood microbiota.

## 2. Methods

### 2.1. Participants, Training Protocol and Sample Collection

With the approval by the Ethics Committee from the Capital Region of Denmark (reference H-1-2013-064) and informed consent from all participants, eight healthy males, aged between 19 and 27 years old took part in the study. The participants represent a subset of a larger cohort previously described [30,31,32]. Participants followed an approved exercise protocol described by Fabre et al. [32]. Briefly, they performed a 6-week endurance exercise program consisting of 60-min cycling sessions at 70% of their initial VO_2_ max, five days per week. Skeletal muscle biopsies (vastus lateralis) from 8 participants and blood samples from 6 of them were collected under fasting conditions at rest (basal) before the training period and five days after it, to avoid any effects due to a single exercise bout. Samples were instantly snap-frozen in liquid nitrogen and stored at −80 °C for further analysis. The clinical characteristics of the participants are shown in Table 1.

### 2.2. Bacterial DNA Extraction

Total bacterial DNA was extracted as previously described [24]. Bacterial DNA was then sequenced using next generation high throughput sequencing of variable regions of the 16S rRNA bacterial gene, with a specific protocol established, as described (www.vaiomer.com (accessed on 19 February 2021)).

### 2.3. Negative Controls

Negative controls were introduced during DNA extraction and amplification as previously described [33]. Empty tubes collected contaminants from labware and reagents during the extraction steps (Muscle-EXT-NC, Blood-EXT-NC). Similarly, a mock quantitative polymerase chain reaction (qPCR) reaction was performed, lacking DNA from the samples (qPCR-NC). As with muscle and blood samples, negative controls were sequenced, and the resulting amplicons were pooled with amplicons from the samples to create operational taxonomic units (OTUs) (see Clustering).

### 2.4. 16S rRNA Gene Amplicon Sequencing

The V3-V4 region of the bacterial 16S ribosomal gene was amplified by PCR using *Vaiomer* (Vaiomer, Labège, France) universal primers. The resulting amplicons were sequenced using Illumina (Illumina Inc., San Diego, CA, USA) 2 × 300 paired-end MiSeq technology to encompass 476 base pairs. Amplicon sequences either shorter than 350 nucleotides (nt), longer than 480 nt, or without the two primers, allowing for 10% mismatch, were removed as well as sequences with at least one ambiguous nucleotide (N).

### 2.5. Clustering

The read sequences from samples and negative controls were clustered by similarity into OTU Operational Taxonomic Units using the swarm algorithm v2.1.6 [34]. We performed the clustering in two steps, the first using an aggregation distance of 1, the second with an aggregation distance of 3. OTUs identified as chimaeras by the ‘vsearch’ v1.9.5 software [35] were removed together with OTUs with an abundance lower than 0.005% of the whole dataset. OTUs were assigned taxonomy by sequence alignment to sequences in the databank RDP v11.4 (https://rdp.cme.msu.edu/ (accessed on 19 February 2021)) using Blast+ v2.2.30+ [36]. OTUs with coverage and identity ≥80% to the phiX174 phage (NC_001422.1) sequence, used as the internal control in Illumina sequencing, were removed. The process produced 289 OTUs, and we recorded for each sample the number of PCR amplicons belonging to each OTU. The sample was the tissue, either blood or muscle, taken from one participant either before or after training, e.g., participant11_muscle_pretraining; resulting in 28 samples = muscle (8 participants × 2 training status) + blood (6 participants × 2 status). Each element y_gi_ of the resulting count matrix constituted the number of PCR amplicons from sample *i* belonging to OTU g. The number of OTU- and sample-specific PCR amplicons over the number of sample-specific PCR amplicons represented the OTU relative abundance.

### 2.6. Comparing Taxa Abundance before and after Training

We ran the linear discriminant analysis effect size (LEfSe) algorithm [37] on the entire set of OTUs to determine the clades most likely to explain the difference between before and after training in either tissue. Differential taxa abundance was calculated on the filtered count matrix using edgeR v2.7.0d [38,39]. The edgeR software fits a negative binomial distribution to replicated count-based data. Biological replicates were the participants who took part in this study and the counts, the number of PCR amplicons belonging to each OTU in each sample. We used the model~0 + group, where the group was a factor encoding the tissue (muscle or blood) and the training status (pretraining, posttraining), e.g., muscle_pretraining. OTUs were filtered using edgeR filterByExpr with default parameters. We calculated differential taxa abundance using the edgeR quasi-likelihood test on the contrasts muscle_posttraining–muscle_pretraining and blood_posttraining–blood_pretraining.

### 2.7. Correlation between OTU Counts and Clinical Parameters

We calculated the Spearman correlation between the combination of values listed below, and we adjusted the *p*-values for multiple testing corrections using the Benjamini–Hochberg method [40]. We ran the first four correlations for both tissues, whilst the last for muscle only.

Alpha distributions vs. clinical parametersΔ (posttraining-pretraining) alpha distribution vs. Δ (posttraining-pretraining) clinical parametersFiltered OTUs counts vs. clinical parametersΔ posttraining-pretraining) filtered OTUs counts vs. Δ (posttraining-pretraining) clinical parameters.

### 2.8. Visualisation

Plots were generated using the ggplot2 R package [41].

## 3. Results

### 3.1. Bacterial DNA Is Present in Human Muscle and Blood

To determine if bacterial DNA is present in healthy human tissues, we amplified and sequenced the V3-V4 variable regions of 16S rRNA obtained from aseptically extracted skeletal muscle and blood samples of 8 healthy males. We detected bacterial 16S rRNA in all samples, with no significant difference in 16S copy numbers per ng of extracted DNA before and after training within each tissue (Appendix A). Bacterial DNA content in blood was significantly lower than in muscle: on average, 340 fewer 16S copy numbers per ng, p_adj_ = 2.5 × 10^−6^, (Figure 1). We measured possible bacterial DNA contamination in various negative control samples and found extremely low levels of bacterial DNA concentrations in controls versus experimental samples (Appendix A and Figure 1; p_adj_ < 0.001). More specifically, we detected that the buffer used for muscle DNA extraction (Muscle-EXT-NC) contained 13 times less bacterial DNA than muscle samples and that the buffer used for blood DNA extraction (Blood-EXT-NC) had 12 times less bacterial DNA than blood samples. Moreover, amplification of the blank qPCR showed 36 and 27 times less bacterial DNA than muscle and blood samples, respectively. Multidimensional scaling plot based on the OTU highest abundances shows marked segregation between negative controls and experimental samples, except BlcExt-AP1 and BlcExp-AP2, for which a marginal overlap with muscle samples is detected (Appendix A). These results indicate that contamination is low in amount and also, given the difference like bacterial DNA between background noise and experimental samples, that bacterial DNA contamination was very unlikely to be a bias in the detection of resident bacterial DNA in muscle and blood.

Phyla distribution in the entire cohort and sample-wise analysis of Operational Taxonomic Units (OTUs) showed that Proteobacteria, Actinobacteria, Firmicutes and Bacteroidetes account for the major phyla in both muscle and blood (Figure 2). Few OTUs did not show any similarity to known bacteria (labelled as unknown) or were ambiguously interchangeable under different phyla (labelled as multi-affiliation). Negative controls had greater diversity in represented phyla and an increased Actinobacteria’s and Bacteroidetes’s relative abundances compared to the samples (Figure 2A). Alpha diversity, a measure of bacteria diversity within each sample, showed similar diversity across groups before and after endurance training (Appendix A). Collectively, these results demonstrate that diverse types of bacterial DNA are present in blood and muscle from healthy humans.

### 3.2. Exercise Remodels Bacterial DNA Content in the Skeletal Muscle, but Not in the Blood

Next, we aimed to determine if physical exercise training influences the composition of tissue bacteria. We detected differences in the bacterial population in muscle but not in blood (Figure 3A,B). Differential abundance analysis confirmed the significant difference in skeletal muscle microbiota with 2 *Pseudomonas* OTUs being less abundant, and both *Staphylococcus* and *Acinetobacter* being more abundant after exercise (Figure 3C).

In blood, we detected a single unknown genus of Proteobacteria phylum being significantly less abundant after exercise along with a *Corynebacterium* OTU (Figure 3D). In contrast, another *Corynebacterium* OTU increased after exercise, showing that the microbial composition was globally unchanged (Figure 3D).

Linear discriminant analysis effect size (LEfSe) supported the diminished abundance of Pseudomonadaceae, *Pseudomonas* and the increase of Moraxellaceae, *Acinetobacter* in skeletal muscle after training (Figure 4C) and resulted in a decrease of *Burkholderiales* and increase of *Bacteroidales* and Clostridiales in muscle after training (Figure 4A). In blood, LEfSe analysis showed an increase in *Micrococcaceae* after training (Figure 4B,D).

### 3.3. No Correlation between OTUs Profiles and Clinical Parameters

To gain insight into the potential physiological function of tissue-borne bacteria, we searched for any association between OTUs counts and the clinical parameters of each participant (Table 1). We found no association between OTUs profiles in blood or muscle and clinical parameters such as anthropometric measures, fasting insulin and glucose, triglycerides and cholesterol levels and aerobic capacity.

To get insight into the potential influence of training efficiency on bacterial profiles, we analysed the link between improved aerobic capacity (ΔVO_2_max), classified into high, medium or low aerobic capacity improvement (Appendix A), and the bacterial composition in blood and skeletal muscle. The principal coordinate analysis (PCoA) plot of Jaccard distances did not show any grouping of samples with similar training efficiency (Appendix A). In the PCoA, the lack of grouping of samples with an equal increment of aerobic capacity reflects a lack of association between the increased capacity of the heart to deliver blood to the muscle and bacterial profiles.

## 4. Discussion

Here, we detected a substantial amount of bacterial DNA in skeletal muscle and blood from young healthy humans. Using bacterial DNA as a surrogate marker of the presence of bacteria, we analysed the bacterial composition of blood and skeletal muscle in healthy men before and after endurance training. We discovered that the composition of bacteria is different after training in muscle but not in blood, supporting exercise may induce a specific remodelling of the skeletal muscle microbiome.

Bacterial translocation to peripheral tissues are the subject of intense debate, and concerns about potential contamination have shed doubt on bacterial DNA’s actual presence in tissues [33]. Contaminating DNA may swamp the low-biomass amount of bacterial material in tissues, potentially leading to false-positive results [42]. Our laboratories have vast experience in conducting low-biomass sample handling with particular caution to prevent contamination [3,16,24,25,26,43,44]. To control for possible contamination, we meticulously controlled for contamination at two major steps of sample handling: at the DNA extraction level and during PCR amplification. Quantification of background noise levels showed several logs of magnitude between the background noise and experimental samples, and a marked difference in the bacterial origin of extracted DNA in background versus tissue samples. This indicates that contamination of bacterial DNA in our experimental procedures is unlikely to have biased the detection of resident bacterial DNA in tissues.

In this study, we did not try to characterise the absolute levels of bacterial communities, but instead comparatively identified tissue-specific changes resulting from endurance training. We argue that the confounding and taxon-specific contamination is unlikely to favour a sample over another and represents a low-level noise that is evenly present in all samples. Differences between samples, which we identified by rigorous statistical analysis, are unlikely to be attributed to contamination but rather to a genuine change in the tissue-borne microbial composition. Moreover, it is important to stress that the relative abundances are shown in Figure 3A,B do not account for the taxa’s absolute quantities. Total bacterial DNA concentrations measured by qPCR, as shown in Figure 1, help to better appreciate the low magnitude of noise levels. This is particularly relevant when comparing tissue samples and negative controls, as the qPCR amplification showed up to 13 times fewer copies of the 16S rRNA gene than the former.

Using the same methodology for the analysis of bacterial DNA as in this study, Lluch et al. have readily detected bacterial DNA in a variety of mouse tissues, including the brain, muscle, adipose tissue, liver and heart [25]. In humans, tissue-borne bacterial DNA profiling and the association with tissue-specific pathologies has been performed for instance, by associating bacterial signatures to different malignant histological grades of human breast tissue [45]. Yet, in these studies like in our study, the legitimate question regarding the actual presence of live bacteria remains to be investigated, as the bacterial DNA that we and others detect in mouse and human tissues may originate from dead bacteria or intracellular bacteria after phagocytosis. A communication showing electronic microscopy images of intact bacteria in the human postmortem brain suggests that alive bacteria are physiologically present in healthy tissues [46]. Recent results demonstrate as well the presence of bacteria in adipose depots [2]. While electron microscopy could validate the presence of intact bacteria, our study suggests a potential role of skeletal muscle bacteria in muscle biology regulation.

The bacterial DNA that we detected in blood and skeletal muscle belongs primarily to the Proteobacteria phylum, followed by Actinobacteria, Firmicutes and Bacteroidetes phyla. Of interest, these phyla represent the main bacterial bulk in the human gut, though in very different frequencies, since Firmicutes and Bacteroidetes represent 90% of the gut bacterial community. Actinobacteria, Proteobacteria and Fusobacteria are represented at subdominant levels in the human gut and are highly variable among individuals [47,48]. We also detected Chlamydia, Candidatus saccharibacteria and Fusobacteria phyla. Chlamydia has been previously detected in the human stomach [49], Candidatus saccharibacteria in the oral cavity [50] and Fusobacteria in the oral cavity and gastrointestinal tract [51]. In skeletal muscle and blood, the detection of DNA that belongs to bacteria which typically populate the digestive tract suggests migration of bacteria or bacterial DNA from the gut.

Increased intestinal permeability (IP) results from a loss of tight junction integrity in people undergoing physiological stress, such as that produced by strenuous exercise [52,53,54]. During intense and prolonged physical activity, the body temperature increases (hyperthermia), the system produces stress hormones [55] and reactive oxygen species (ROS) [56], and blood flows away from the gastrointestinal (GI) tract towards muscles, heart and lungs [57]. Hyperthermia and blood redistribution cause intestinal barrier disruption [58] allowing for bacterial components to transfer outside the GI tract and trigger immune and inflammatory responses [59]. An increased presence of pro-inflammatory cytokines, such as tumour necrosis factor-alpha (TNFα), interferon-alpha (IFNα), interferon-gamma (INFγ) and interleukins (IL1β or IK6), increases the opening of the intestinal epithelial tight junctions thereby exacerbating the intestinal permeability [60]. Interestingly, IP due to splanchnic hypoperfusion has been observed in healthy men already after 1 h of endurance exercise at 70% of maximum aerobic workload capacity [61]. IP is known to increase in T2D patients, giving rise to a persistent chronic, low-grade inflammation, leading to the onset of insulin resistance [62] and autoimmune disorders [63]. Data show that regular exercise reduces IP, thereby breaking the vicious circle of chronic inflammation and improving glucose metabolism [62,63].

In this study, participants trained daily at moderate intensity (one hour per day, 5 days a week for 6 weeks at 70% of their maximum aerobic capacity), which may enhance the intestinal barrier and tight-junction integrity through short-chain fatty acid (SCFA) produced by symbiotic bacteria [64]. As an alternative to passive translocation through a permeable intestinal barrier, bacteria may translocate via an active migration process. Accordingly, an active mechanism of translocation mediated by endocytosis was previously described [65]. Epithelial cells continuously ingest viable bacteria, which pass from the intestine lumen, through the cell, to the lamina propria. There, bacteria are met by macrophages which engulf and kill them and potentially transport them to peripheral sites through the bloodstream [65]. Active transportation by intestinal macrophages would also explain the absence of inflammatory responses towards bacterial DNA found in the blood and the muscle of the participants. Indeed, intestinal macrophages do not secrete inflammatory cytokines and promote inflammatory anergy towards indigenous bacteria [66]. During post-thymic education of the immune system, intestinal macrophages distribute bacterial debris to peripheric organs. In this way, macrophages educate regulatory T cells and shape the T-cell receptor (TCR) repertoire to accommodate antigens deriving from commensal microbiota [67].

Of interest, we found that exercise induces remodelling of bacterial DNA in skeletal muscle, but not in blood. Although these changes are intriguing, the change in muscle microbiome may be a consequence of increased blood flow to exercising skeletal muscles, which may favour the infiltration of aerobic bacteria such as *Acinetobacter*, replacing less O_2_-dependent bacteria such as *Pseudomonas*. Macrophage infiltration in muscle tissue was described as a mechanism of repair and regeneration upon endurance exercise training [68], which may account for an active and regulated translocation of bacterial DNA to the tissue. The very adaptive nature of skeletal muscle may also play a role in the difference in bacterial DNA between muscle and blood, as six weeks of exercise training induces profound remodelling of the skeletal muscle tissue that is still present after a 5-day recovery, while most blood parameters return to baseline levels. Besides, at the level of the gut barrier or at the tissue level, investigations tracing bacterial migration may shed light on the mechanisms by which the DNA of specific bacteria populate skeletal muscle and what are their physiological triggers.

Our attempt to link bacterial DNA profiles in blood and muscle and some clinical parameters of each respective participant did not return and significant association. There are several potential explanations for these negative results. Notably, we measured clinical parameters at the steady-state, where they may have returned at a baseline level. In this case, the clinical variables would not correlate with the posttraining bacterial DNA profiles. This hypothesis is particularly relevant in exercise training studies since the molecular triggers of adaptations to exercise are quickly back to basal levels days after the last exercise bout. For instance, the elevation of baseline levels of circulating glucose and lipids, which participates in changing gene expression in response to exercise, is transient. Another possible explanation for the lack of association between clinical parameters and bacterial DNA content is the small number of variables we tested and the relatively small sample size of our study cohorts, which does not allow in-depth multi-regression analyses.

Understanding how tissues respond to bacterial DNA fragments from specific bacteria would provide great insight into the possible regulatory role of bacteria, or at least their DNA, on skeletal muscle function. Evidence supports that bacterial DNA binds to Toll-like Receptor 9 (TLR9), which in turn controls numerous immune cell functions [69]. Among the known Toll-like Receptors, TLR9 appears to be the only subtype able to detect DNA from self and non-self [69]. While to our knowledge, no studies have specifically investigated if bacterial DNA binds to TLR9 in skeletal muscle, it has been shown that TLR9 exerts cellular protection in cardiomyocytes [70]. In cardiomyocytes, a CpG-oligodeoxynucleotide was shown to have the potential to bind temporally to TLR9 and to reduce the use of energy substrates, thereby activating AMP-activated protein kinase (AMPK) and protecting the cardiomyocyte [70]. This action was exerted without inducing canonical inflammatory signalling, suggesting that extracellular DNA released from damaged tissue or bacteria is interpreted as a sign of danger by the cell [71]. Bacterial DNA could therefore be considered as a triggering signal of danger, to which the skeletal muscle cell could adapt by modifying energy metabolism, as previously reported [72]. Such mechanism could be at play in exercised muscle under specific stress conditions, although further studies are warranted.

Finally, in the present study, we have analysed tissues solely from healthy young men. However, results from other types of subjects (e.g., females, elders, individuals with obesity or type 2 diabetes) may differ and need investigation.

## 5. Conclusions

We demonstrate here that bacterial DNA is present in blood and muscle from healthy young men, and we provide evidence that endurance exercise can specifically remodel bacterial DNA in skeletal muscle. Our study makes ground for further investigations aiming to determine the contribution of skeletal muscle bacteria on muscle function.

## Figures and Tables

**Figure 1 biomedicines-10-00064-f001:**
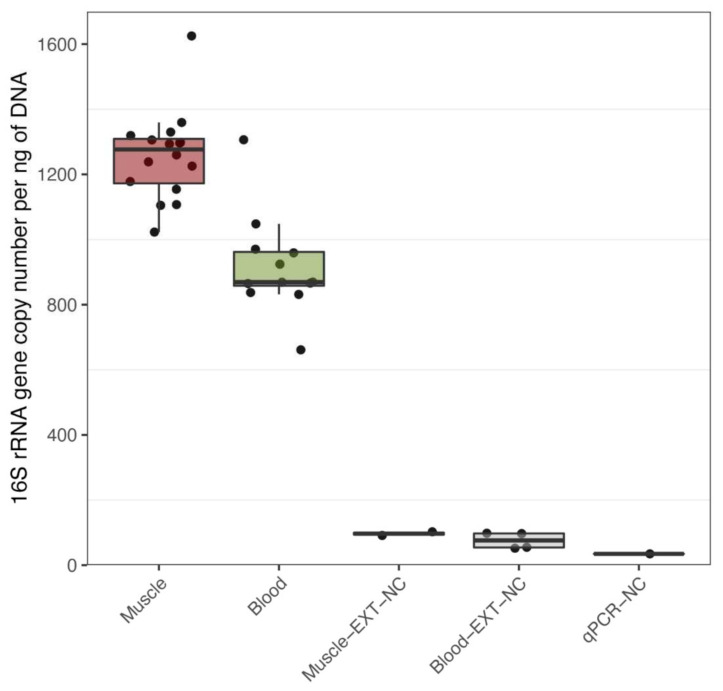
Quantitative PCR detection of bacterial DNA in muscle and blood. Quantification of 16s rRNA per ng of total DNA as measured by quantitative polymerase chain reaction (qPCR) in skeletal muscle biopsies (Muscle), blood (Blood), as well as in the following negative controls (NC) buffers used of DNA extraction for skeletal muscle (Muscle-EXT-NC), blood (Blood-EXT-NC) and water, buffers and reagents used for PCR (qPCR-NC). Analysed by Tukey’s multiple comparison test using 95% family-wise confidence level.

**Figure 2 biomedicines-10-00064-f002:**
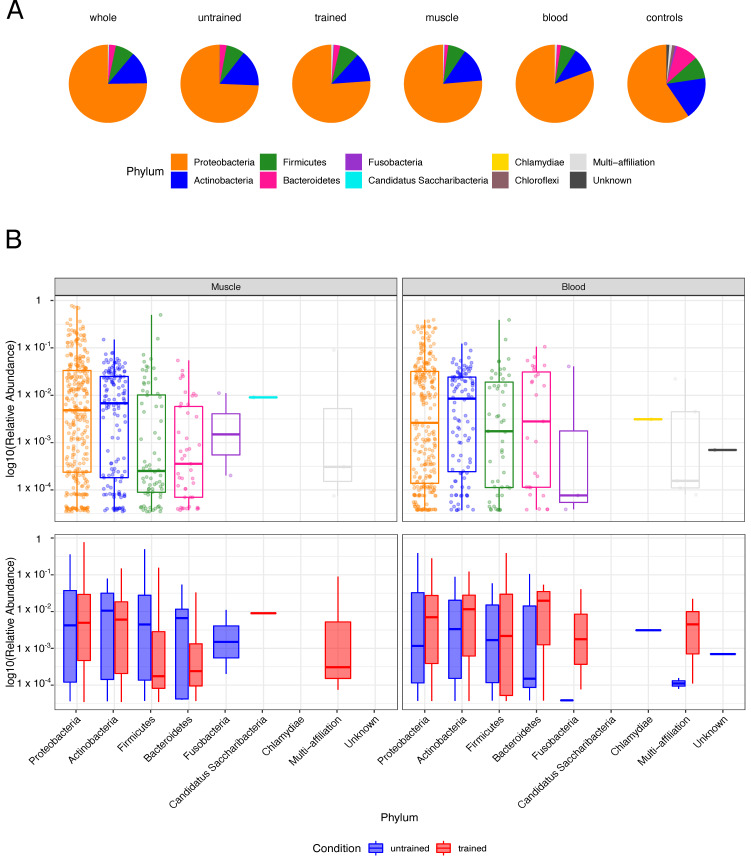
Phyla composition in tissues and controls: (**A**) The pie charts depict the phyla distributions in the entire cohort (whole), samples of untrained and trained participants, muscle and blood samples and negative controls (controls). (**B**) The boxplots display the Log10-transformed, sample-wise relative abundances of operational taxonomic units (OTUs) by phylum for the entire cohort (top) as well as before and after training (bottom).

**Figure 3 biomedicines-10-00064-f003:**
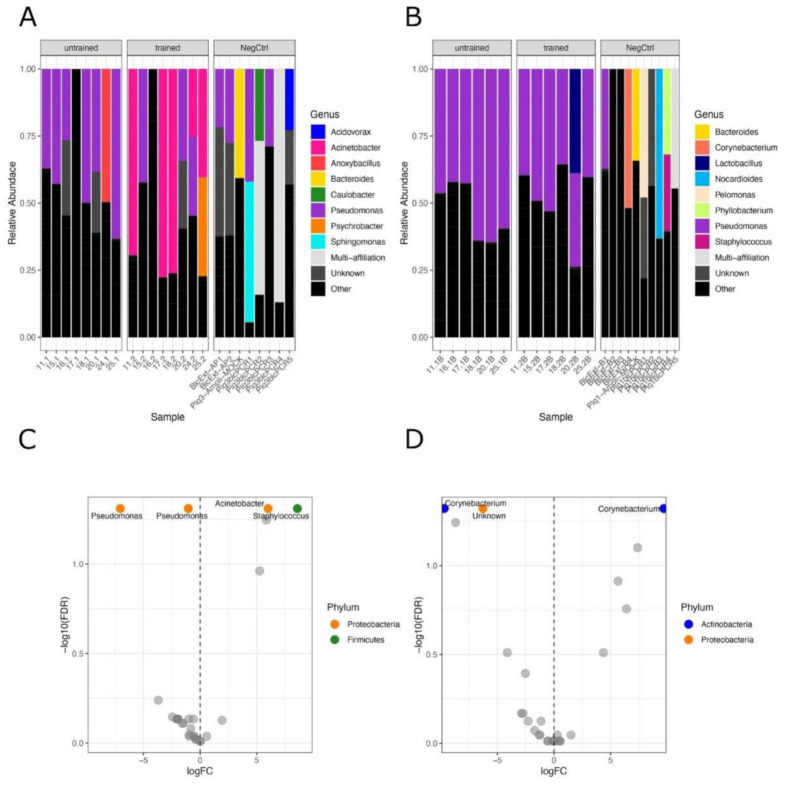
Phylogenetic distribution of bacteria in tissues before and after training: Stacked bar plots showing the phylogenetic distribution of bacteria at the genus level before and after training across participants in muscle (**A**) and in the blood (**B**). The NegCtrl panels show the phyla relative abundances of concomitant controls samples. C and D, Volcano plots displaying the differentially abundant genera before and after exercise in muscle (**C**) and blood (**D**), obtained by statistical differential abundance analysis performed with edgeR. Labelled OTUs are significant ones, FDR < 0.05.

**Figure 4 biomedicines-10-00064-f004:**
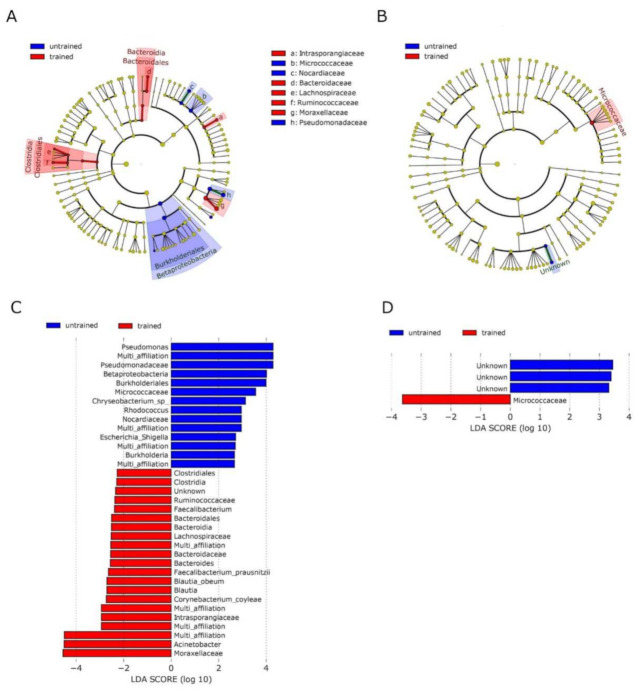
Analysis of taxonomic differences after training in blood and muscle: Linear discriminant analysis effect size (LEfSe) was used to identify taxonomic differences in the microbiota of skeletal muscle (**A**,**C**) and blood (**B**,**D**) before and after exercise training. Cladograms A and B give a representation of relevant OTUs on the taxonomic tree. C and D show ranked differential OTUs by effect size. The input file for the LEfSe analysis was obtained from the ‘phyloseq’ object using the LEfSe function of yungtools2 package. The LEfSe docker container (https://hub.docker.com/r/biobakery/lefse (accessed on 19 February 2021)) was used to perform LEfSe analysis and figure generation.

**Table 1 biomedicines-10-00064-t001:** Clinical characteristics of study participants.

	Untrained, *n* = 8	Trained, *n* = 8
Age—years	24 ± 4	24 ± 4
Weight—kg	79.4 ± 10.0	78.5 ± 9.2
Body mass index—kg/m^2^	22.89 ± 2.21	22.71 ± 2.08
Waist—cm	87 ± 7	80 ± 7 ***
Hip—cm	94 ± 5	90 ± 5 **
Waist/Hip	0.93 ± 0.04	0.89 ± 0.05 *
VO_2_—mL	3694 ± 514	4332 ± 458 ***
VO_2_/kg	46.5 ± 4.4	55.4 ± 4.3 ***
Glucose (fasting)—mmol/L	4.9 ± 0.4	5.1 ± 0.4
Insulin—pmol/L	64 ± 24	64 ± 35
HOMA-IR	2.31 ± 0.94	2.46 ± 1.56
HbA1c—%	34 ± 3	33 ± 3
Plasma cholesterol (total)—mmol/L	1.3 ± 0.8	4.4 ± 0.4
Low-density lipoprotein—mmol/L	6.7 ± 0.9	2.7 ± 0.4
High-density lipoprotein—mmol/L	1.2 ± 0.3	1.3 ± 0.2 *
Triglyceride—mmol/L	4.6 ± 0.8	1.2 ± 0.3
C-reactive protein—mg/L	1.4 ± 0.2	1.0 ± 0.0
Leukocytes—×10^9^/L	2.7 ± 0.6	6.1 ± 1.1

Data are presented ±SD. *** indicates *p* < 0.001; ** *p* < 0.01; * *p* < 0.05. Homeostatic Model Assessment for Insulin Resistance (HOMA-IR); Hemoglobin A1c (HbA1c).

## Data Availability

The data presented in this study are available on request from the corresponding author. The data are not publicly available due to sensible data involving humans.

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
