# Peer review of "Endurance Training in Humans Modulates the Bacterial DNA Signature of Skeletal Muscle"

_biomedicines, 2021, doi:10.3390/biomedicines10010064_

Round 1
Reviewer 1 Report
Dear author,
Thank you for your contribution. Although the article sounds good, I see it will be better if you have a separate section for the literature review as the article has not analysed enough previous related studies. providing such analysis would provide evidence of the novelty of the article.
Best regards
Author Response
We thank the reviewer for this comment. We appreciate the suggestion of the reviewer to expand on the literature to give the reader a more comprehensive understanding of the state of the art in the field. However, as the field of tissue microbiome is only emerging, we do not identify significantly more literature than what we already describe in the introduction of our manuscript. Regardless, as this article is an original research article, we are constrained by the editorial space and we would not be able to include a separate, literature review section to our manuscript.
Reviewer 2 Report
This study by Vollarroel and colleagues investigated bacterial profiles of human skeletal muscle and blood of healthy young males following a 6-week endurance training protocol along with determining the capacity/influence of 'background' contamination. Their results showed that the influence of background contamination was much lower in control samples compared to tissue. Interestingly, and opposite to the findings from the gut microbiota, the Proteobacteria, Actinobacteria, Firmicutes and Bacteroidetes phyla were the most abundant in skeletal muscle muscle and blood samples. Finally, the 6-week exercise training stimulus induced more pronounced effects on esident bacterial DNA in skeletal muscle but not in blood. These results from a prominent research team are highly novel within the field of processes mediating human skeletal muscle adaptation to exercise and advance current knowledge for endurance training to remodel bacterial DNA in skeletal muscle. The paper is very well written and my comments are minor in nature.
1) What were you hypothesizing? Please include an appropriate hypothesis in the introduction as this helps with providing direction to the information provided in the introduction
2) Regarding methods: How were the processes of DNA extraction and amplification performed in samples obtained?
3) Regarding methods: Has the process of collecting contaminants from labware and reagents during the extraction steps been used/published in the literature before? Please cite these references and/or provide further basis to the basis of this approach.
4) Lines 172-173: Would it not be more accurate to conclude that multiple diverse types of bacterial DNA are present in blood and muscle from healthy humans?
5) Lines 326-335: It it possible the remodelling being alluded to here is due to the fact that by 6 weeks of exercise training, skeletal muscle tissue is in an active adaptation process (in this case, enhanced substrate metabolism and mitochondrial biogenesis due to endurance training), compared to the circulatory system which, after a 5-day recovery from the last exercise bout, is unlikely to be perturbed.
Author Response
We thank the reviewer for these comments, that we address in a point-by-point fashion below:
1) What were you hypothesizing? Please include an appropriate hypothesis in the introduction as this helps with providing direction to the information provided in the introduction
Numerous studies demonstrate that the gut microbiota changes in response to exercise. We have shown earlier that the gut microbiota can translocate to tissues, notably the adipose depots, leading to a change in tissue function such as adipogenesis. Therefore, a change in muscle microbiota could occur following exercise and contribute to muscle metabolism as triggered by exercise.
We have now added the following hypothesis to the introduction: "Here, we hypothesised that a lifestyle intervention like physical exercise training remodels the composition of blood and skeletal muscle-borne bacteria”
2) Regarding methods: How were the processes of DNA extraction and amplification performed in samples obtained?
We apologize for omitting this important information. We have now amended the methods section with the following text:
Total bacterial DNA was extracted as previously described [65]. Bacterial DNA was then sequenced using next generation high throughput sequencing of variable regions of the 16S rRNA bacterial gene, with a specific protocol established, as described (www.vaiomer.com).
[65]. Valle C, Servant F, Courtney M, Burcelin R, Amar J, Lelouvier B. Comprehensive description of blood microbiome from healthy donors assessed by 16S targeted metagenomic sequencing. Transfusion. 2016, 56(5):1138-1147.
3) Regarding methods: Has the process of collecting contaminants from labware and reagents during the extraction steps been used/published in the literature before? Please cite these references and/or provide further basis to the basis of this approach.
Thank you. We have added reference “Schierwagen R, Alvarez-Silva C, Servant F, Trebicka J, Lelouvier B, Arumugam M: Trust is good, control is better: technical considerations in blood microbiome analysis. Gut 2020, 69(7):1362-1363.” to the methods section.
4) Lines 172-173: Would it not be more accurate to conclude that multiple diverse types of bacterial DNA are present in blood and muscle from healthy humans?
Thank you for the suggestion. We have replaced “Collectively, these results demonstrate that bacterial DNA is present in blood and muscle from healthy humans.” (line 172-173) by “Collectively, these results demonstrate that multiple diverse types of bacterial DNA are present in blood and muscle from healthy humans.”
5) Lines 326-335: It it possible the remodelling being alluded to here is due to the fact that by 6 weeks of exercise training, skeletal muscle tissue is in an active adaptation process (in this case, enhanced substrate metabolism and mitochondrial biogenesis due to endurance training), compared to the circulatory system which, after a 5-day recovery from the last exercise bout, is unlikely to be perturbed.
Thank for this interesting insight. We have added the sentence “The very adaptive nature of skeletal muscle may also play the difference in bacterial DNA between muscle and blood, as six weeks of exercise training induces profound remodelling of the skeletal muscle tissue that is still present after 5-day recovery while most blood parameters return to baseline levels.” to this part of the discussion.Reviewer 3 Report
The authors investigated influences of 6-week endurance training on bacterial DNA contents and profiles in muscle and blood. The authors seemed to perform experiment carefully to avoid contamination. They discussed potential contamination bias well. However, the significance of changes in bacterial DNA profiles in blood and muscles in response to physical training were not explained enough. Therefore, I am afraid that the importance of findings in the present study might not be understood well by readers.
Major comments
Overall points.
In introduction, the significance and importance of investigating changes in bacterial DNA content in blood and tissues particularly in skeletal muscle were not explained.
In addition, I want to know how bacterial DNA profiles would affect functions of skeletal muscle and other tissues.
Abstract
In line 23-24, the authors described “endurance training specifically remodels the bacterial DNA profile of skeletal muscle in humans.” However, in the present study, characterization of participants was quite restricted (e.g. no female, no aged people). Therefore, I recommend to conclude like “endurance training specifically remodels the bacterial DNA profile of skeletal muscle in young healthy males”. (also in Conclusions)
Methods
In the present study, the authors measured bacterial DNA levels in tissues 5 days after the final training. Why did you choose this time point? Is there possibility that changes in bacterial DNA levels in response to physical training was underestimated due to the sampling time point?
Results
Please revise the position of contents in table 1.
Discussion
I recommend to describe your opinion on possible differences in results obtained from other populations (e.g. females, aged subjects, obese subjects, diabetics etc.)
Author Response
We thank the reviewer for her/his comments and questions. We hereby address each point individually.
Major comments
Overall points.
In introduction, the significance and importance of investigating changes in bacterial DNA content in blood and tissues particularly in skeletal muscle were not explained.
Thank you for this comment. We have clarified on the significance of investigating skeletal muscle tissue in the introduction. We have added “To assess the effect of endurance training on bacterial signatures in human peripheral tissues, we opted to sequence bacterial DNA content from skeletal muscle, a primary exercise-effector tissue, and in blood, as surrogate markers of the possible presence of tissue-borne bacteria.”
In addition, I want to know how bacterial DNA profiles would affect functions of skeletal muscle and other tissues.
We understand and appreciate the reviewer's interest in the potential effects of bacterial DNA profiles in the function of skeletal muscle and other tissues. While we share the same interest, we currently lack experimental evidence investigating the possible function of bacterial DNA in skeletal muscle.
The bacterial translocation and hence evidence of tissue microbiota is not new and has been reported previously (Berg RD, Wommack E, Deitch EA (1988) Immunosuppression and intestinal bacterial overgrowth synergistically promote bacterial translocation. Arch Surg 123 (11):1359-1364). While bacterial DNA is likely a biomarker of the presence of bacteria in tissues, we cannot yet discriminate whether such bacteria are alive.
As described in the discussion, we can only speculate that tissue-borne bacterial DNA may serve as a signalling mechanism informing distally of the status of the gut homeostasis. Metagenomic studies revealing bacterial genes coupled with tissue transcriptomics may shed light on the environmental impact of the microbial community on the metabolic functions of skeletal muscle and other tissues.
Abstract
In line 23-24, the authors described “endurance training specifically remodels the bacterial DNA profile of skeletal muscle in humans.” However, in the present study, characterization of participants was quite restricted (e.g. no female, no aged people). Therefore, I recommend to conclude like “endurance training specifically remodels the bacterial DNA profile of skeletal muscle in young healthy males”. (also in Conclusions)
The reviewer raises a very valid point. We have corrected the sentences in the abstract and in the conclusion and added this statement to the discussion to enhance clarity: “In the present study, we have analysed tissues solely from healthy young men. However, results from other types of subjects (e.g., females, elders, individuals with obesity or type 2 diabetes) may differ and need investigation.”.
Methods
In the present study, the authors measured bacterial DNA levels in tissues 5 days after the final training. Why did you choose this time point? Is there possibility that changes in bacterial DNA levels in response to physical training was underestimated due to the sampling time point?
Thank you. The reason for sampling after five days from the final training was to avoid effects linked to the single exercise-bout and to attribute results solely to the enduring aspect of the training. We agree with the reviewer that within the 5-day recovery period, the bacterial signature in blood has had the time to repristinate its original state and showed to be unperturbed. We have now reviewed and addressed this point together with the differences between the tissues in the following piece of the discussion:
“The very adaptive nature of skeletal muscle may also play the difference in bacterial DNA between muscle and blood, as six weeks of exercise training induces profound remodelling of the skeletal muscle tissue that is still present after 5-day recovery while most blood parameters return to baseline levels.”
Results
Please revise the position of contents in table 1.
Thank you. We have now fixed the table.
Discussion
I recommend to describe your opinion on possible differences in results obtained from other populations (e.g. females, aged subjects, obese subjects, diabetics etc.)
Thank you for this suggestion. Without experimental evidence in other populations, it is difficult, at this stage, to elaborate further on the nature of the difference that we may observe in other populations. We have expressed carefulness in the generalisation of our findings to all populations in the following sentence added to the end of the discussion: “Finally, in the present study, we have analysed tissues solely from healthy young men. However, results from other types of subjects (e.g., females, elders, individuals with obesity or type 2 diabetes) may differ and need investigation.”.
Round 2
Reviewer 3 Report
Thank you for reply. The authors sincerely answered my questions. But I have one more request relating to my previous comment (one of overall points). I would like to request your consideration on that point.
I have understood that the possible function of bacterial DNA in skeletal muscle has not been elucidated. But in introduction (perhaps also in discussion), if there are appropriate previous research that implies actions of bacterial DNA (or bacteria per se) accumulated in circulation and/or deep tissues, I recommend showing possible influences of alternations in bacterial DNA profiles on physiological functions of tissues (or whole body) (e.g. metabolism, gene expression, inflammation etc.). It would help readers to understand scientific significance and importance of the present analysis. Most of readers would have interest on what would happen if bacterial DNA profiles have altered in blood and tissues.
For example, ref. 16 (Anhe et al. 2020) showed a certain degree of basis for analyzing microbial profile found in plasma, liver and adipose tissue of obese subjects in the manuscript (e.g. “Gut bacteria and their fragments have been shown to translocate beyond the intestinal barrier, colonise and/or accumulate in the blood and extra-intestinal tissues, and trigger immunogenic pathways that can affect glucose homeostasis and other cardiometabolic outcomes.” in introduction.).
Author Response
We agree with this reviewer that addressing the impact of bacterial DNA on muscle physiology is an important point to be addressed. We have included this paragraph to the discussion:
“Understanding how tissues respond to bacterial DNA fragments from specific bacteria would provide great insight into the possible regulatory role of bacteria, or at least their DNA, on skeletal muscle function. Evidence supports that bacterial DNA bind to Toll-like Receptor 9 (TLR9), which in turn controls numerous immune cell function [74]. Among the known Toll-like Receptors, TLR9 appears to be the only subtype able to detect DNA from self and non-self [74]. While to our knowledge, no studies have specifically investigated if bacterial DNA binds to TLR9 in skeletal muscle, it has been shown that TLR9 exerts cellular protection in cardiomyocytes [75]. In cardiomyocytes, a CpG-oligodeoxynucleotide was shown to have the potential to bind temporally to TLR9 and to reduce the use of energy substrates, thereby activating AMP-activated protein kinase (AMPK) and protecting the cardiomyocyte [75]. This action was exerted without inducing canonical inflammatory signalling, suggesting that extracellular DNA released from damaged tissue or from bacteria is interpreted as a sign of danger by the cell [76]. Bacterial DNA could therefore be considered as a triggering signal of danger, to which the skeletal muscle cell could adapt by modifying energy metabolism, as previously reported [77]. Such mechanism could be at play in exercised muscle under specific stress conditions, although further studies are warranted.”
[74] Haas T, Metzger J, Schmitz F, Heit A, Müller T, Latz E, Wagner H. The DNA sugar backbone 2' deoxyribose determines toll-like receptor 9 activation. Immunity. 2008 Mar;28(3):315-23. doi: 10.1016/j.immuni.2008.01.013. PMID: 18342006.
[75] Shintani Y, Kapoor A, Kaneko M, Smolenski RT, D'Acquisto F, Coppen SR, Harada-Shoji N, Lee HJ, Thiemermann C, Takashima S, Yashiro K, Suzuki K. TLR9 mediates cellular protection by modulating energy metabolism in cardiomyocytes and neurons. Proc Natl Acad Sci U S A. 2013 Mar 26;110(13):5109-14. doi: 10.1073/pnas.1219243110. Epub 2013 Mar 11. PMID: 23479602; PMCID: PMC3612600.
[76] Matzinger P. Tolerance, danger, and the extended family. Annu Rev Immunol. 1994;12:991-1045. doi: 10.1146/annurev.iy.12.040194.005015. PMID: 8011301.
[77] Shintani Y, Drexler HC, Kioka H, Terracciano CM, Coppen SR, Imamura H, Akao M, Nakai J, Wheeler AP, Higo S, Nakayama H, Takashima S, Yashiro K, Suzuki K. Toll-like receptor 9 protects non-immune cells from stress by modulating mitochondrial ATP synthesis through the inhibition of SERCA2. EMBO Rep. 2014 Apr;15(4):438-45. doi: 10.1002/embr.201337945. Epub 2014 Mar 7. PMID: 24610369; PMCID: PMC3989675.